# EEG and Physiological Signals Dataset from Participants during Traditional and Partially Immersive Learning Experiences in Humanities

Rebeca Romo-De León [1,†], Mei Li L. Cham-Pérez [1,†], Verónica Andrea Elizondo-Villegas [1,†], Alejandro Villarreal-Villarreal [1,†], Alexandro Antonio Ortiz-Espinoza [2], Carol Stefany Vélez-Saboyá [2], Jorge de Jesús Lozoya-Santos [1] , Manuel Cebral-Loureda [2] and Mauricio A. Ramírez-Moreno [1,*]

[1] Mechatronics Department, School of Engineering and Sciences, Tecnologico de Monterrey, Monterrey 64849, Mexico; a01282945@tec.mx (R.R.-D.L.); a01139386@tec.mx (M.L.L.C.-P.); a01720357@tec.mx (V.A.E.-V.); a01570397@tec.mx (A.V.-V.); jorge.lozoya@tec.mx (J.d.J.L.-S.)

[2] School of Humanities and Education, Tecnologico de Monterrey, Monterrey 64849, Mexico; alexandro.ortiz@tec.mx (A.A.O.-E.); carol.velez@tec.mx (C.S.V.-S.); manuel.cebral@tec.mx (M.C.-L.)

[*] Correspondence: mauricio.ramirezm@tec.mx

[†] These authors contributed equally to this work.

**Abstract:** The relevance of the interaction between Humanities-enhanced learning using immersive environments and simultaneous physiological signal analysis contributes to the development of Neurohumanities and advancements in applications of Digital Humanities. The present dataset consists of recordings from 24 participants divided in two groups (12 participants in each group) engaging in simulated learning scenarios, traditional learning, and partially immersive learning experiences. Data recordings from each participant contain recordings of physiological signals and psychometric data collected from applied questionnaires. Physiological signals include electroencephalography, real-time engagement and emotion recognition calculation by a Python EEG acquisition code, head acceleration, electrodermal activity, blood volume pressure, inter-beat interval, and temperature. Before the acquisition of physiological signals, participants were asked to fill out the General Health Questionnaire and Trait Meta-Mood Scale. In between recording sessions, participants were asked to fill out Likert-scale questionnaires regarding their experience and a Self-Assessment Manikin. At the end of the recording session, participants filled out the ITC Sense of Presence Inventory questionnaire for user experience. The dataset can be used to explore differences in physiological patterns observed between different learning modalities in the Humanities.

**Dataset:** The data generated from this experimentation are available for download on Figshare https://doi.org/10.6084/m9.figshare.24777084, accessed on 28 January 2024.

**Dataset License:** The dataset is made available under the Creative Commons 4.0 (CC BY) license.

**Keywords:** neuroeducation; neurohumanities; immersive spaces; neuroengineering; educational innovation

## 1. Summary

Recently, digital technology has become involved in higher education to potentially improve the student experience, considering student involvement and engagement, and enhance learning [1]. Some of the most used technologies in education are virtual reality and immersive spaces [2]. Most of the research conducted on the use of educational technology has focused on the engineering and science fields, and few studies incorporate the use of these tools in the Humanities field, including subjects such as human society, culture, politics, language, literature, history, art, and social psychology [3]. This represents an area of opportunity to improve students' experience and learning and overcome the

limitations of a subjective evaluation through Digital Humanities [1]. In this paper, a dataset is described. Twenty-four participants were recruited and divided into two groups (twelve participants in each group) that simulated two learning scenarios. The control group participated in a learning experience in a traditional classroom setup, while the experimental group participated in a partially immersive environment, both in a simulated Humanities class addressing the topic of *The Passions of Descartes* [4]. The partially immersive environment was designed by the Neurohumanities Lab from Tecnologico de Monterrey university and is further described below. This environment combined technology and algorithms for real-time biometric data acquisition, such as electroencephalography (EEG), electrodermal activity (EDA), blood volume pressure (BVP), inter-beat interval (IBI), and engagement (from EEG signals); artificial intelligence (AI) models for emotion, motion, and voice recognition; AI text-generators, speech-to-text converters, computer vision-based facial gesture recognition, and image-to-image AI image generator; and feedback loops used to track physiological signal changes that modify the classroom environment (e.g., changes in visualizations, lighting, and sound).

The database was acquired over one week (20–27 October 2023), during which both scenarios were tested. Within the two groups, EEG and three-dimensional head acceleration signals (ACCs) were acquired using an OpenBCI Ultracortex Mark IV headset (OpenBCI, New York, NY, USA); BVP, IBI, EDA, and wrist temperature (TEMP) signals were acquired using an Empatica E4 (Empatica, Milano, Italy) wearable bracelet. During the experiments, questionnaires were applied to collect psychometric data from participants. These questionnaires included the General Health Questionnaire (GHQ), Trait Meta-Mood Scale (TMMS-24), ITC-Sense of Presence Inventory (ITC-SOPI) presence questionnaire, and the Self-Assessment Manikin (SAM) test, which evaluates the level of valence, arousal, and dominance (VAD).

The learning experience (experiment) was divided into four different blocks, which are referred to as "scenes", each consisting of different specific tasks. Participants in both groups took part in all of the four scenes (which had a different setup between groups, but the same objective). The design of the scenes was based on Kant's philosophy on the Theory of Knowledge following a structure of cognitive learning and integrative levels, which will be further described in Section 3.2.

For all participants (in both groups), physiological signals were captured during baseline states, as well as during the four scenes. The physiological signals and the questionnaires' information mentioned beforehand in this dataset will allow researchers to explore whether changes in the perceived levels of presence in both experiences reflect differences between physiological signals' patterns. This dataset can be of particular interest to educators in the Humanities since it has been reported that higher levels of presence and engagement in immersive spaces are linked to optimized learning [1,5].

- **Subject:** Biomedical Engineering, Neuroengineering.
- **Specific subject area:** EEG acquisition, Presence, Humanities.
- **Type of data:** Physiological signals (EEG, BVP, IBI, EDA, TEMP, head ACC, emotion recognition and engagement values), and psychometric data (GHQ, TMMS-24, SAM, and ITC-SOPI questionnaires).
- **How data were acquired:**
  - EEG and ACC signals were acquired using the OpenBCI Ultracortex Mark IV headset with eight dry electrodes (FP1, FP2, C3, C4, P7, P8, O1, and O2) located over the scalp based on the 10-20 system, and two ear clips on both earlobes. Head ACC was obtained from an integrated accelerometer into the headset, containing three-dimensional (XYZ) acceleration.
    - ∗ **Sampling rate:** 250 Hz.
    - ∗ **Reference:** Both earlobes.
  - BVP, EDA, IBI and TEMP signals were acquired by wristband Empatica E4 on the subject's dominant hand.

- ∗ **Sampling rate for BVP:** 64 Hz [6].
- ∗ **Sampling rate for EDA:** 4 Hz [6].
- ∗ **Sampling rate for IBI:** dependent on BVP signal quality due to motion [6].
- ∗ **Sampling rate for Temperature:** 1 Hz [6].
- – **Demographic data:** 24 participants (10 male and 14 female) between 18 and 25 years old.
- – Screening instruments for psychometric data were acquired by a scaled version of GHQ, and self-reported measures of perceived emotional intelligence were acquired by TMMS-24 before starting the experiment. The SAM test was used after every individual scene to obtain valence, arousal, and dominance data from user experience. [7] Finally, the user's experience of the digital media was acquired by the ITC-SOPI test after the conclusion of the experiment [8].

- **Data format:**
  - – Physiological data from OpenBCI and Empatica E4 devices were recorded and organized into five files: one resting state file consisting of physiological signals of the participants and four files recorded during users' experiences during each of the scenes.

- **Description of data collection:**
  - – Twenty-four participants (twelve in the control group and twelve in the experimental group) performed four different scenes (educational tasks in the Humanities field context). In the experimental group, the participants were able to interact with the Neurohumanities Lab, a partially immersive environment for completing tasks. Meanwhile, the control group completed the tasks on paper in a classroom-like scenario.

- **Data source location:**
  - – **Institution:** Tecnológico de Monterrey.
  - – **City/State:** Monterrey/Nuevo León.
  - – **Country:** México.

## 2. Data Description

This dataset contains physiological signal data and psychometric data from questionnaires collected from 24 participants in two different learning scenarios. The physiological signals include EEG, head ACC data, real-time engagement, and the emotion detected calculated by a custom Python code [9], as well as BVP, EDA, IBI, and TEMP. Each physiological signal was recorded for each participant in a resting state and in the four different scenes that comprise the experiment. Files were saved with the name SXXRYY, where XX stands for participant ID (01–24) and YY stands for repetition or scene number (00 being resting state recording and 01–04 being the four scenes in chronological order). Participant IDs 01–12 were participants in the control group and participant IDs 13–24 were in the experimental group. An overview of all folders and files in the dataset is shown in Figure 1.

There are two zip files for the EEG Headset data recorded from OpenBCI, "RawEEG.zip" (containing EEG and head ACC data) and "EmotionsCSVRaw.zip" (containing a CSV file that includes real-time engagement and emotions detected), each with 120 files in CSV format according to resting state and learning scene (24 participants × 5 scenes). For the Empatica data, there are 120 folders (one for each subject and each scene), and within the main folder of each subject, there are three folders, where the raw folder contains all physiological signals (BVP, EDA, IBI, and TEMP) in CSV files. In total, for the data recorded from Empatica E4, there are 480 files.

The psychometric data for each participant were collected with different questionnaires during the recording sessions, including the GHQ, TMMS-24, SAM, and ITC-SOPI. All answers were collected and summarized into one single XLSX file. Demographic data were also stored for each participant in a single XLSX file. Impedance values from the Open BCI EEG Headset are also summarized for each participant and reported in a single XLSX file

inside the "EmotionsCSVRaw.zip". In summary, this dataset contains a total of 723 files in either a CSV or XLSX file format arranged in different folders and zip files for an easier download.

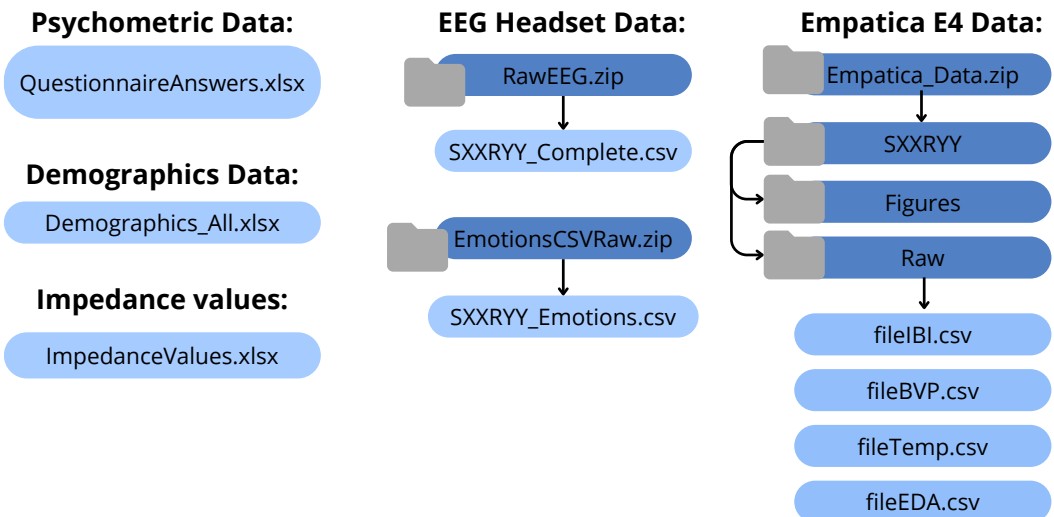

**Figure 1.** Overview of folders and files available for download in Figshare. XX: Participant number ranging from 1 to 24. YY: Repetition (scene) number ranging from 0 to 4. The Figures folder is an empty folder created automatically by the code used for Empatica E4 data acquisition.

### 2.1. Demographics

A total of 24 Spanish-speaking participants (10 f., 14 m., aged $21.3333 \pm 1.4043$) were recruited as volunteers. The following inclusion criteria were considered for this study: healthy adults aged between 18 and 25 years, not having a mental-related illness diagnosis, and not receiving pharmaceutical treatment for any mental illnesses.

For every participant, age, gender, and handedness were reported in one single (.xlsx) file called Demographics_All.xlsx, and its content is described in Table 1.

**Table 1.** Overview of data stored in Demographics_All.xlsx.

| Category | Column | File Name |
|----------|--------|-----------|
| Participant ID | 1 | Demographics_All.csv |
| Group | 2 | Demographics_All.csv |
| Gender | 3 | Demographics_All.csv |
| Handedness | 4 | Demographics_All.csv |

### 2.2. Psychometric Data: Questionnaires

Psychometric data from participants were collected through different questionnaires across the experimental session, and the answers were stored in Google Forms. It is important to note that all questionnaires were applied in Spanish since all participants were Spanish-speakers living in Monterrey, Mexico (at the time of the experiment). All answers were compiled in a single Microsoft Excel (.xlsx) document called QuestionnaireAnswers.xlsx, and each questionnaire was divided into one Excel sheet. GHQ and TMMS-24 questionnaires were applied at the beginning of each session, while the SAM Questionnaire was applied between each of the scenes, including additional questions in a 5-point Likert scale that were created by the research team for further understanding of the users' experience. At the end of the session, the ITC-SOPI questionnaire was applied. All questionnaires were applied to all participants in both groups. In the following list, each questionnaire objective is further described.

- **General Health Questionnaire**
    - GHQ is a twelve-question multiple-choice questionnaire used to obtain qualitative data from the participants; parameters include: degree of health, sleep quality, and emotional health [10].
- **Trait-Meta Mood Scale TMMS-24**
    - This questionnaire was applied at the beginning of the experimentation to obtain data about the participants' emotional intelligence [11].
- **Self-Assessment Manikin test**
    - The SAM test is a non-verbal and pictorial questionnaire where participants evaluate their affective response to specific stimuli. In this case, this test was applied between scenes during the experience using a 5-point Likert scale, where 1 is the lowest stimuli and 5 the highest stimuli for each of the categories [12]. According to the VAD model, valence describes how pleasantly a stimulus is felt, arousal relates to the intensity of how an emotion is felt, and dominance reflects the degree of control the participant has over their own affective state and emotions [7,12]. This test was used in both control and experimental group to compare and contrast the valence, arousal, and dominance level of participants.
- **ITC-SOPI**
    - ITC-SOPI was used to evaluate the participants' presence in four different categories: spatial presence, engagement, ecological validity, and negative effect [8]. This questionnaire was answered by participants at the end of the four scenes and comprises a total of 32 questions,; each question is evaluated with a 5-point Likert scale, where 1 means totally disagree and 5 means totally agree) [8].

### *2.3. Physiological Signals*

Physiological data from OpenBCI and Empatica E4 devices were recorded and organized into five files per subject: one resting state file and four files recorded during users' experience in each of the four scenes.

### 2.3.1. Electroencephalography and Head Acceleration

Raw EEG signals were recorded with custom Python codes [13] with a sampling rate of 250 Hz. Both earlobes were used as reference, and eight channels (Fp1, Fp2, C3, C4, P7, P8, O1, O2) according to the 10-20 International System were recorded, as depicted in Figure 2.

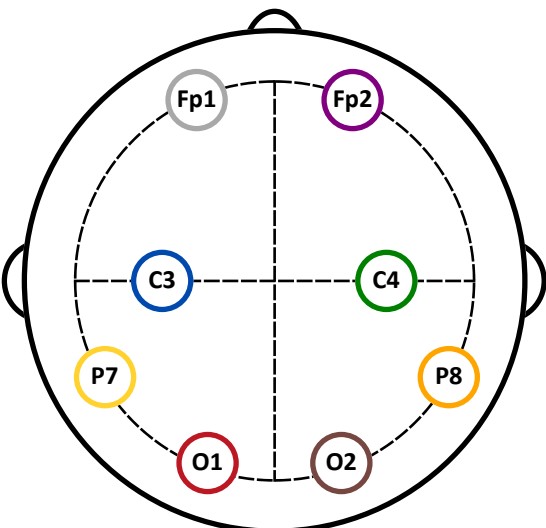

**Figure 2.** Distribution of the eight electrodes used for EEG acquisition in this database.

The raw EEG data of all subjects are located in the *RawEEG.zip* folder. In Table 2, an overview of the files' contents is shown. Each file contains 13 columns. Column 1 contains the sample number, column 2 contains the timestamps in UNIX format, and columns 3–10 contain the EEG samples collected during the timestamps in column 2 following the previously stated electrode order. Columns 11–13 contain the three-dimensional head acceleration signals (XYZ, respectively). For each participant, there is one file which belongs to the (2 min) resting state recording (eyes closed and eyes open, 1 min each) and four files which correspond to the each of the four scenes.

**Table 2.** Overview of EEG data file locations and file name logic.

| Scene | File Name |
| --- | --- |
| Resting state | SXX*R00_Complete.csv |
| Scene 1 | SXX*R01_Complete.csv |
| Scene 2 | SXX*R02_Complete.csv |
| Scene 3 | SXX*R03_Complete.csv |
| Scene 4 | SXX*R04_Complete.csv |

XX*: Participant number, from 1 to 24.

Impedance values are also reported in the file named ImpedanceValues.xlsx. In column 1, subjects are listed, and columns 2–9 contain the impedance values measured in each channel before starting the session recordings. The process to measure impedance values is described in Section 3.1.1. In Figure 3, a histogram of the impedance values of all channels and all subjects is displayed. It can be observed in the histogram that there is a higher frequency of EEG channels with lower impedances, showing that despite using dry electrodes, impedances were reduced as much as possible during the setup process.

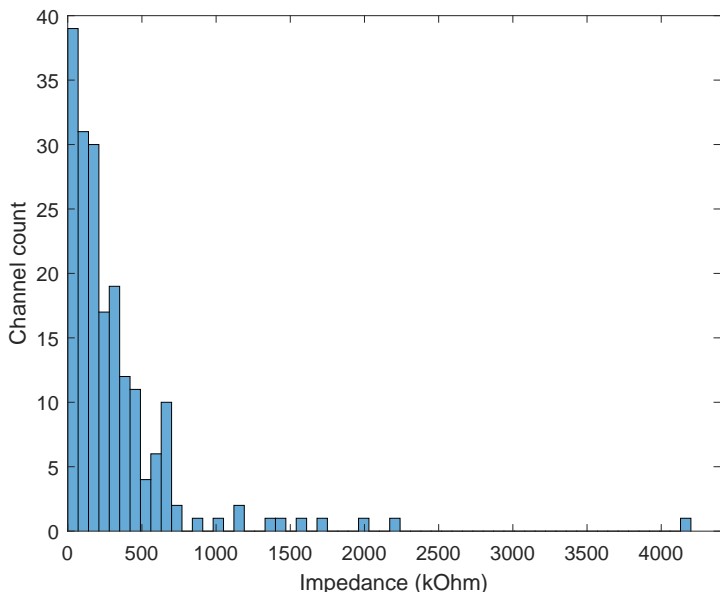

**Figure 3.** Histogram of dry electrode impedance values registered for all 24 subjects and 8 channels.

Emotion recognition and engagement were also calculated every ten seconds in real time and saved on a Comma Separated Value (.csv) file for further analysis. The distribution of these files is described in Table 3. These calculations were made inside a previously coded Python script with trained machine learning (ML) algorithms, which were also used for the adaptation of the Neurohumanities Lab environment [9]. The algorithm used EEG features to estimate engagement in real-time and VAD values every ten seconds and assign to each time window the closest emotion (joy, sadness, desire, admiration, hate, and love, from *Passions of Descartes*) [4] within the three-dimensional VAD space [9]. The engagement was also calculated every ten seconds, and was limited to a 0–1 range. In this

file, there are seven columns containing (1) the number of times an emotion was calculated (time windows), (2) the timestamps in UNIX format, (3) the engagement, (4) the estimated emotion, and (5) the valence, (6) arousal, and (7) dominance values.

**Table 3.** Overview of EEG data file location and file name logic.

| Scene | File Name |
|---|---|
| Resting state | SXX*R00_Emotion.csv |
| Scene 1 | SXX*R01_Emotion.csv |
| Scene 2 | SXX*R02_Emotion.csv |
| Scene 3 | SXX*R03_Emotion.csv |
| Scene 4 | SXX*R04_Emotion.csv |

XX*: Participant number, from 1 to 24.

2.3.2. BVP, EDA, IBI, and TEMP

The following files were generated after recording with the Empatica E4 and a Python [13] script. Raw Empatica signals for BVP, EDA, and TEMP were recorded with Python with a sampling rate of 64 Hz, 4 Hz, and 1 Hz, respectively [6]. The sampling rate for IBI signal was dependent on BVP signal quality due to motion. These data are located in the Empatica_Data.zip folder. Each participant has a folder named "SXXRYY", where "XX" symbolizes the subject number and "YY" symbolizes the scene number. Each raw file in the Raw Folder contains two columns: column 1 contains the timestamps in datetime format (e.g., dd/mm/yy hh:mm:ss.ms) and column 2 contains the signal amplitude for each signal (BVP, EDA, IBI, and temperature) collected during the timestamps in column 1, following the previously stated order. The datetime Excel format was then converted to UNIX timestamp format or time-lapsed in seconds since 1 January 1970 using the following formula in MS Excel =TODAY()-DATE(1970,1,1)$\times$ 86,400 (this last number is the number of seconds in a day). Table 4 shows the file names generated per scene and for each (raw) physiological signal obtained with the Empatica E4.

**Table 4.** Overview of BVP, EDA, IBI, and TEMP data file location and file name logic.

| Scene | BVP | EDA | IBI | Temperature |
|---|---|---|---|---|
| Resting state | SXX*R00_fileBVP.csv | SXX*R00_fileEDA.csv | SXX*R00_fileIBI.csv | SXX*R00_fileTemp.csv |
| Scene 1 | SXX*R01_fileBVP.csv | SXX*R01_fileEDA.csv | SXX*R01_fileIBI.csv | SXX*R01_fileTemp.csv |
| Scene 2 | SXX*R02_fileBVP.csv | SXX*R02_fileEDA.csv | SXX*R02_fileIBI.csv | SXX*R02_fileTemp.csv |
| Scene 3 | SXX*R03_fileBVP.csv | SXX*R03_fileEDA.csv | SXX*R03_fileIBI.csv | SXX*R03_fileTemp.csv |
| Scene 4 | SXX*R04_fileBVP.csv | SXX*R04_fileEDA.csv | SXX*R04_fileIBI.csv | SXX*R04_fileTemp.csv |

XX*: Participant number, from 1 to 24.

## 3. Methods

### 3.1. Experimental Set-Up

3.1.1. EEG-Headset Recordings

EEG and ACC signals were acquired as described in Section 2.3.1. For data acquisition (EEG and ACC), the headset was connected to a laptop using a wireless dongle. A custom Python code collected the data at a 250 Hz sampling rate, including UNIX timestamps for each sample and real-time calculations of engagement and emotion recognition, using a previously trained Machine Learning model [9]. For engagement, the engagement index [7] was estimated every ten seconds, using Equation (1).

$$Engagement\ index = \frac{Beta}{Alpha + Theta} \tag{1}$$

where *Theta*, *Alpha*, and *Beta* represent the average (FP1 and FP2 electrodes) power spectral density (PSD) of the evaluated time windows in the 4–7, 8–12, and 13–30 Hz frequency ranges, respectively [9].

To place the OpenBCI headset over the participant's head, the distance from the nasion to the inion on the sagittal plane was measured, and a mark was traced on the midpoint of this distance. Another mark was drawn on the forehead indicating 10% of this distance. The distance between the participant's tragi was also measured, and the intersection was marked, which is where the central electrode (Cz) should be placed on the headset. These measurements helped to adjust the headset in the right position.

The area where each electrode was positioned was cleaned with rubbing alcohol to improve contact with the skin and obtain better recordings. Each electrode was disconnected when adjusting to avoid wire damage. The reference clips were placed on the earlobes, and the impedance of each electrode was measured with the Cyton Signal widget from the OpenBCI_GUI software [14]. According to the software's recommendation on their systems' impedance values, impedances are considered "good" (below 750 kΩ), "regular" (between 750 and 2500 kΩ), or "bad" (above 2500 kΩ). Impedances were reduced as much as possible during the setup process, but, as can be noted in Figure 3 that most of the times, the electrodes fell within the lower impedance range.

### 3.1.2. Empatica E4 Recordings

Empatica E4 was placed on the dominant hand of the participant. During the positioning of the equipment, an attempt was made to keep the participant as comfortable as possible, avoiding causing any discomfort from the devices. Empatica signals for BVP, EDA, and TEMP were recorded with Python with a sampling rate of 64 Hz, 4 Hz, and 1 Hz, respectively [6]. The sampling rate for IBI signals was dependent on the BVP signals' quality due to motion.

### 3.1.3. Scene Setup

For both groups, the experiment session consisted of four scenes lasting three minutes each. Two different experimental setups were used for both groups, control and experimental. The control group executed all of the tasks in a traditional classroom scenario. Figure 4, shows the setup used during the four scenes of the control group (panels A to D), while panels E to H show the setup applied to the four scenes of the immersive experience.

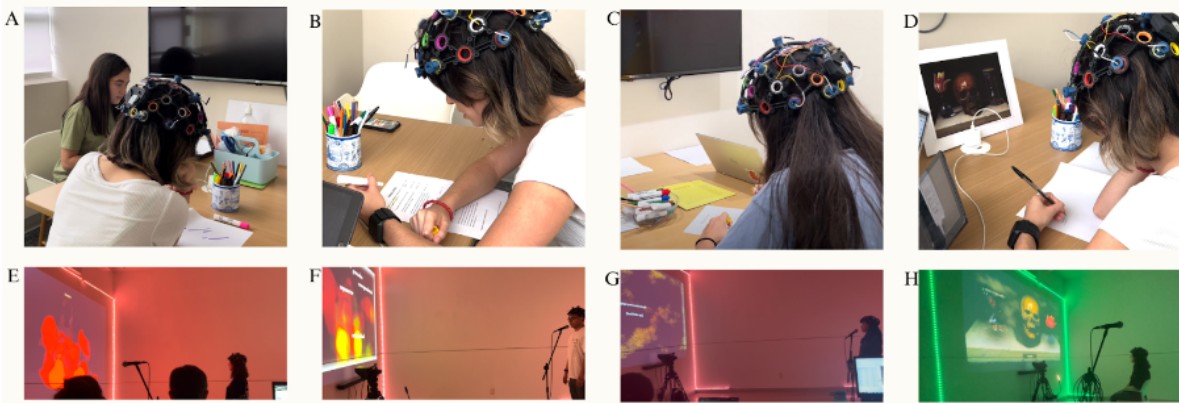

**Figure 4.** Panels (**A–D**) show the experimental setup for the control group for scenes 1, 2, 3, and 4, respectively, while images (**E–H**) show the experimental setup for the experimental group for scenes 1, 2, 3, and 4, respectively.

The following equipment was used for the control group setup in the classroom scenario:

- OpenBCI Ultracortex Mark IV headset;
- Empatica E4 wristband;
- Three Computers (two computers for the experimentation and one to read instructions):
    - PC 1: Used to record OpenBCI's signals and emotion recognition;
    - PC 2: Used to record data from Empatica E4;

- PC 3: Used to read instructions to participants; internal control usage.
- iPad 8th Generation (Apple Inc., Cupertino, CA, USA) was used to fill out the different questionnaires;
- Printed sheets of Descartes' passions;
- Chairs and tables.

The following equipment was used for the experimental group's setup inside the Neurohumanities Lab (Monterrey, Mexico):

- OpenBCI Ultracortex Mark IV headset;
- Empatica E4 wristband;
- SM58 Microphone (Shure, Niles, IL, USA);
- Camera 4K Plus (Blackmagic Studio, Port Melbourne, Australia);
- Five Computers (four to carry out experimentation, one to read instructions to participants):
  - PC 1: Used to record OpenBCI signals and emotion recognition and change room lightning;
  - PC 2: Used to record data from Empatica E4;
  - PC 3: Used for the partially immersive environment (screen projector, sound, and scene creation);
  - PC 4: Used for voice and facial recognition;
  - PC 5: Used to read instructions to participants; internal control usage.
- Epson BrightLink 685Wi+ Screen projector (Seiko Epson Corporation, Tokyo, Japan);
- Professional live stream MX18-Dsklight Lamp;
- Arduino UNO-R3 (Arduino, Italy);
- iPad 8th Generation;
- Printed sheets of Descartes' passions;
- Chairs and tables;
- JBL CINEMA BAR 140 (soundbar and subwoofer) (HARMAN International Industries, Los Angeles, CA, USA).

### 3.2. Experimental Protocol

The experimental protocol was granted an exemption (protocol number EHE-2023-03) by the Institutional Research Ethics Committee (CIEI) of Tecnologico de Monterrey.

As participants arrived, they were asked to sign an informed consent form and then fill out the GHQ and the TMMS-24 questionnaires. After the EEG headset's placement, a 2-minute resting state was recorded. The experimental session consisted of four scenes of three minutes each, where all physiological signal data were recorded and the participant was asked to execute a specific task for each scene. The tasks varied depending on the control and experimental group. The session was based on an educational topic in the Humanities: Descartes' passions [4]. The objective of the tasks was for the participants to gain a better understanding of their emotions through the representation of Descartes' passions.

At the beginning of the session, participants were asked to choose one of six main passions described by Descartes: joy, sadness, desire, admiration, love, and hate [4]. The experimental design, as well as an explanation of each scene, is presented in Figure 5. Even though the two groups involved different setups, the overall tasks (and their objectives) were similar across groups.

The proposed scenes have an underlying logic that provides them with a fundamental structure. These are the different levels of cognition into which modern philosophy has divided experience: scene 1 begins with a physical experience, in which no language is involved, only bodily expression; scene 2 continues with a reflective exercise, in which participants introduce the conceptual explanation of their previous experience; scene 3 involves the previous concepts in articulated sentences, going beyond the conceptual sphere of knowledge to enter the discursive one, where nuances such as meaning and emphasis are taken into account; scene 4 ends with a symbolic synthesis of the whole experience,

appealing to figurative images that can embody different meanings and interpretations of the same fact. Exactly this cognitive structure composed of these different and integrative levels can be observed, for example, in Kant's philosophy, which is the main reference of modern theory of knowledge [15–17].

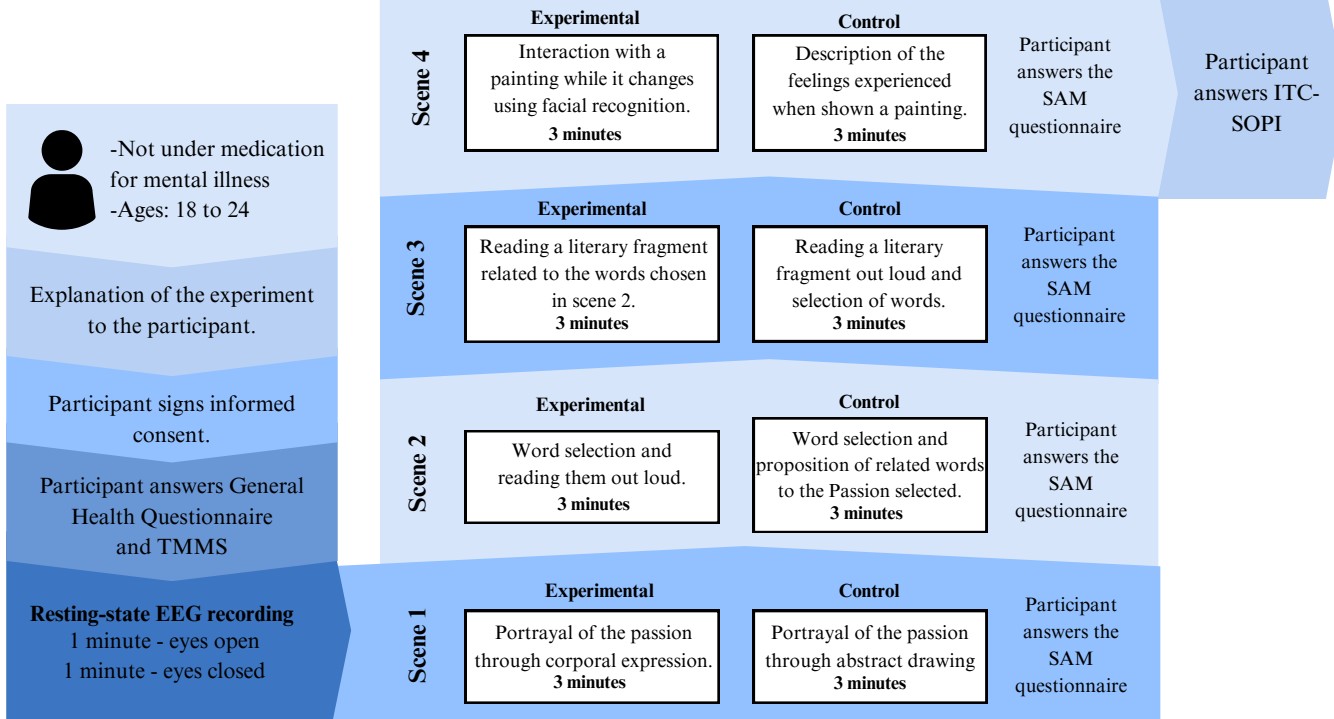

**Figure 5.** Experimental design for control and experimental groups.

### 3.2.1. Description of Scene 1

The first scene aimed for the participant to express the passion chosen, either by body expression or with a handmade drawing. The participants in the experimental group, in the Neurohumanities Lab experience, were told to express their chosen Passion through body movement. They were able to see their movements projected on a screen, which were recorded by a motion detection camera. The detected emotion (related to Descartes's Passion) was shown at the top of the screen, while the chosen passion was shown at the bottom of the screen. Participants were told to try to match the chosen passion with the passion detected by the EEG signal. On the other hand, participants in the control group didn't receive any feedback on emotion detection, nevertheless, they were as well instructed to express their chosen passion by creating an abstract drawing with supplied material, pen, and markers.

It is important to notice that the experimental group executed all tasks inside the Neurohumanities Lab, which received real-time feedback from the participant's physiological signals and changed the sound and lighting in the Lab accordingly. The Descartes' passion detected was only visible for participants in Scene 1, which was the only scene in which the instructions asked them explicitly to match the detected passion with the chosen one. However, the detected emotion file stored as (.csv) is available for all scenes and all recordings, regarding experimental or control group.

### 3.2.2. Description of Scene 2

The second scene asked the participant to choose words from a list that were related to the chosen passion. For the immersive group, the participant's selection was performed by saying the words out loud and virtually adding it to a list on the screen through voice recognition inside the Neurohumanities Lab. The control group selected words by choosing

them from a printed word bank and writing them with a pen on US letter-sized paper in the classroom-like scenario. Different printed word banks were available for each of Descartes' passions.

The words used in the word bank arise from the processing of a corpus of 60 universal literary texts that span from the beginning of the 16th century to the second half of the 20th century, in other words, what can be understood, in a broad way, as the modern era. Examples of authors in this corpus include but are not limited to Russian writers, Edgar Allan Poe, Johann Wolfgang Goethe, and Gabriel García Marquez. This corpus was firstly tokenized, and a word-embedding model was then created.

From this word-embedding model, the word bank was generated by building a natural language processing (NLP) model in Spanish language from the aforementioned corpus. To create this model, a Spanish pipeline from the spacy library ("es_core_news_lg") was used. Additionally, the natural language toolkit (NLTK) [18] and regular expression (RE) libraries [19] were imported as a tokenizer and stopword tool, respectively, to clean the corpus text. Then, the Gensim library [20] was applied using the Word2Vec tool [21] for large corpora to train and model with 100-dimensional Spanish-word-embedding vectors. Once the model was trained, the word bank was created using the emotion words and the most similar function to generate new correlated words.

### 3.2.3. Description of Scene 3

The third scene aimed for the participant to relate the reading of literary fragments with the chosen passion. Inside the partially immersive experience, the literary fragments appeared on the screen and were chosen through artificial intelligence considering the previous word list made by the participant. The participant was asked to read the literary fragments out loud. Meanwhile, the task for the control group consisted of the reading of printed literary fragments on US Letter-sized paper; a different copy of literary fragments was available for each of Descartes' passions.

### 3.2.4. Description of Scene 4

The final scene aimed for the participant to interact or describe a painting, *Vanitas Still Life with a Tulip, Skull and Hour-Glass* by Philippe Champaigne [22]. Inside the Neurohumanities Lab, face recognition was used so that the participants' face and head movements were reflected onto the painting projected on the screen. Additionally, the image-to-image AI generator modifies the flower and hourglass elements based on facial expression detection.

On the other hand, the task for the control group consisted of writing the description and emotions that the painting transmitted to the participant.

Between each scene, an online questionnaire (stored in Google Forms), was distributed to the participants, which they had to answer before starting the next scene. The questionnaire was based on two parts, with three questions based on a Likert scale with possible answers ranging from 1 to 5 points, and the evaluation of the SAM questionnaire (evaluation of valence, arousal, and dominance) according to the experience of the participant in each scene. At the end of the fourth scene, participants were asked to answer the ITC-SOPI questionnaire to evaluate their overall experience. These answers were also stored in Google Forms.

### 3.3. Data Preprocessing

The physiological signals were preprocessed to clean them from noise sources and execute further analysis on patterns related to both groups' experiences.

### 3.3.1. EEG Preprocessing

Figure 6 shows the suggested procedure to preprocess the EEG signals, which allows for obtaining denoised signals. The PREP-Pipeline algorithm was used to remove trends and line noise and to re-reference the signals [23]. A bandpass filter using the EEGLAB default FIR function with a range of frequency from 0.1 to 50 Hz was applied afterwards,

followed by the Artifact Subspace Reconstruction (ASR), $\kappa = 15$, and Independent Component Analysis (ICA) algorithms [24,25]. Additionally, by using EEGLAB's ICLABEL plugin, noise sources were identified and removed to further decontaminate the signals. All of these processing steps were executed using MATLAB 2022a [26].

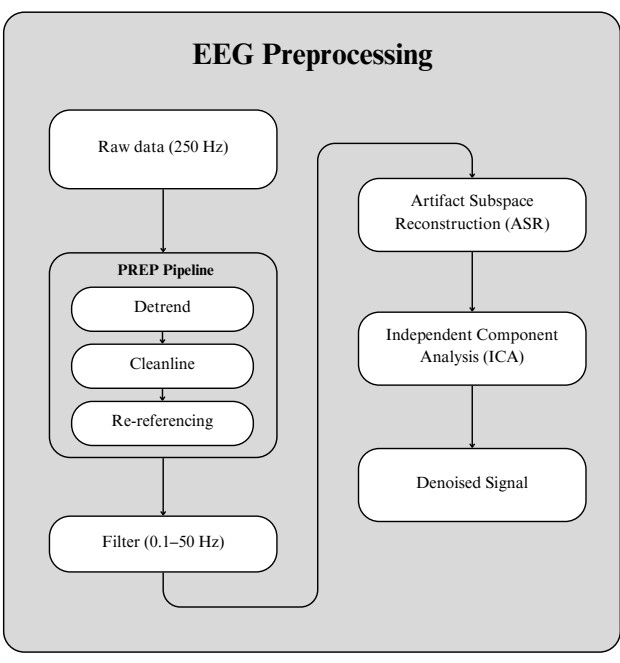

**Figure 6.** Data preprocessing methodology used to analyze EEG data obtained from the experiment.

### 3.3.2. BVP and EDA Preprocessing

Meanwhile, Figure 7 shows the experimental procedure implemented to denoise and analyze the Empatica E4 signals. This procedure includes a synchronization process for EEG data and obtaining the Skin Conductance Response (SCR) component from the EDA and Heart Rate from the BVP signal. All of these processing steps were executed using MATLAB 2022a [26].

The processing methodology starts with the raw data collected from the wristband. These data are filtered natively to reduce motion noise. Then, three different procedures can be implemented with the signals. First, data can be resampled to 250 Hz (upsample) to synchronize Empatica and EEG data and to compare both signals at a given moment. The second process involves the use of a Gaussian filter on the EDA signal, with a 40-point window and a sigma value of 400 ms to obtain the SCR component of the EDA signal. After the filtering process, the SCR can be obtained with MATLAB's Bio-SP toolbox [27,28]. Finally, heart rate can be calculated from the raw BVP signal. MATLAB's findpeaks function can be applied to this signal, and the time span between successive peaks is used to determine the heart rate of each participant.

Figure 8 shows an example of the signals acquired from the experimentation from participant 20, scene 1. Signals shown in the figure include the denoised eight-channel EEG signal and head acceleration data from the OpenBCI headset and resampled raw BVP and EDA signals from the Empatica E4, sharing the same timestamps to allow observation and analysis (if desired) of both EEG and Empatica data in synchrony.

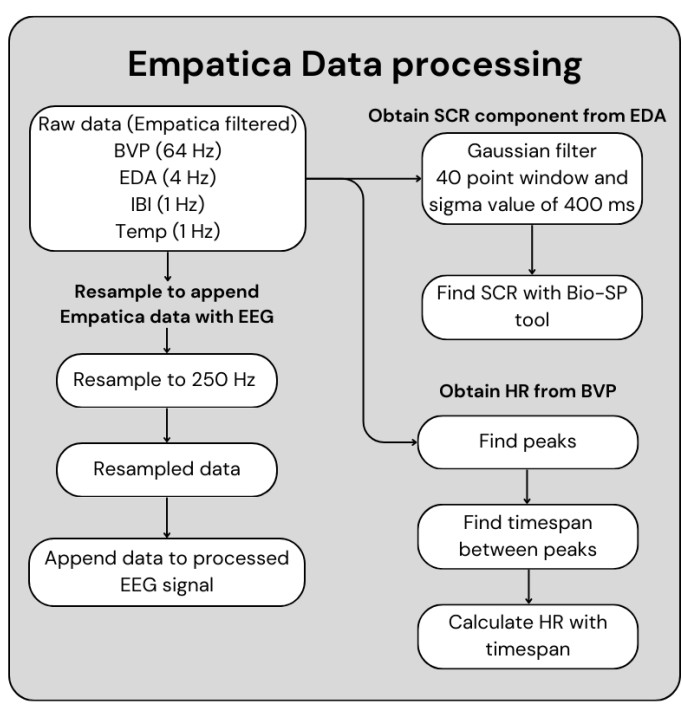

**Figure 7.** Data processing methodology used to analyze heart rate and electrodermal activity data obtained from the experiment.

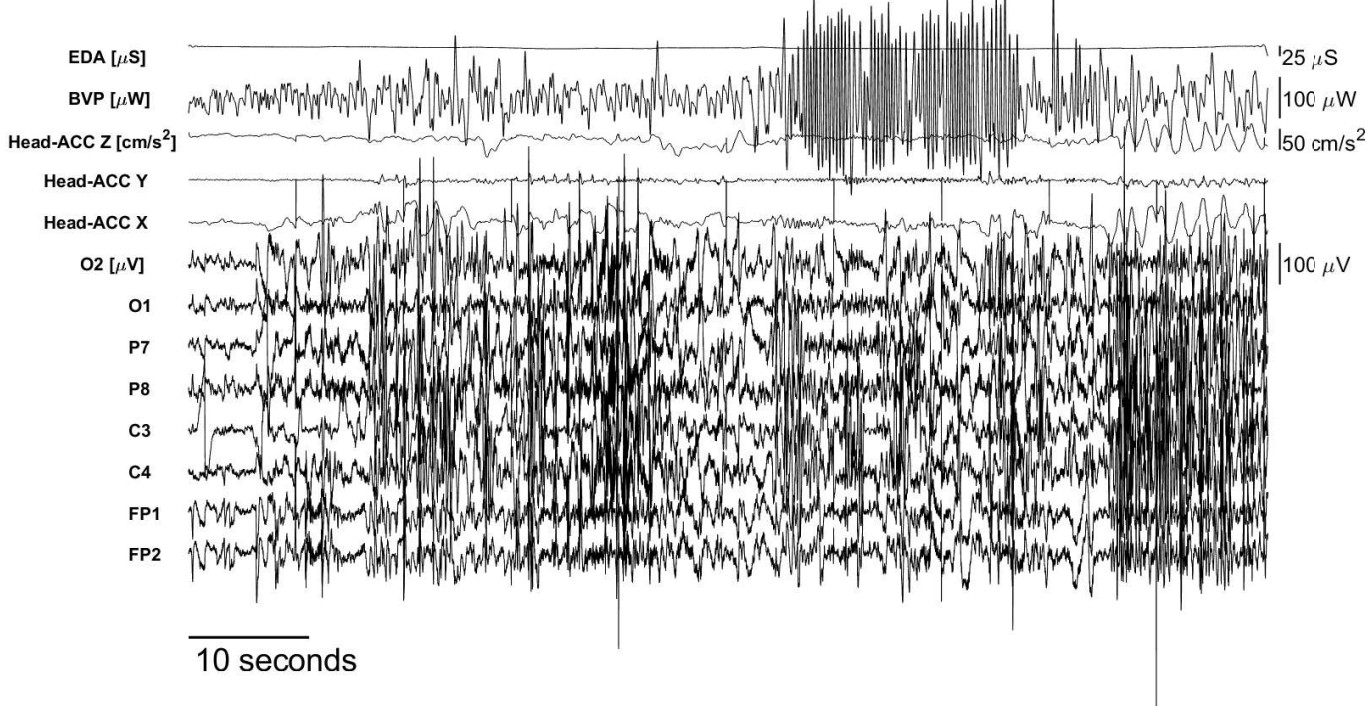

**Figure 8.** EEG, BVP, and EDA signals recovered from partially immersive group subject 20 during the last scene.

## 4. Data Set Limitations and Considerations

While the dataset was carefully created and curated, some limitations were identified and should be considered for further data processing and analysis. First, the sample size is small and limited time-wise to the recordings of 24 participants. It is suggested to execute the same experimental setup with a bigger sample size in the future to be able to generalize

findings. This dataset is also limited in scope, focusing only on the Spanish-speaking population, leaving aside the possible impact on results if this same experimental setup was executed with other populations.

Furthermore, the OpenBCI headset uses dry electrodes which are susceptible to noise and higher impedance levels than wet electrodes, which requires considerable effort during the preprocessing stage before analyzing the dataset. Therefore, a complete preprocessing method is strongly recommended to denoise all physiological signals.

Despite the limitations, the presented dataset contributes to widening and improving learning opportunities in the Humanities field, introducing technology to enhance learning experiences and overcome subjectivity in the evaluation of progress.

**Author Contributions:** Conceptualization, M.C.-L., C.S.V.-S. and M.A.R.-M.; methodology, R.R.-D.L., M.L.L.C.-P., V.A.E.-V., A.V.-V., M.C.-L., A.A.O.-E., C.S.V.-S. and M.A.R.-M.; software, A.A.O.-E., M.C.-L., C.S.V.-S. and M.A.R.-M.; validation, R.R.-D.L., M.L.L.C.-P., V.A.E.-V., A.V.-V., C.S.V.-S. and M.A.R.-M.; formal analysis, R.R.-D.L., M.L.L.C.-P., V.A.E.-V. and A.V.-V.; investigation, R.R.-D.L., M.L.L.C.-P., V.A.E.-V., A.V.-V., C.S.V.-S. and M.A.R.-M.; resources and data curation, R.R.-D.L., M.L.L.C.-P., V.A.E.-V. and A.V.-V.; writing—original draft, R.R.-D.L., M.L.L.C.-P., V.A.E.-V. and A.V.-V.; writing—review and editing, visualization, and supervision, C.S.V.-S., M.A.R.-M. and M.C.-L.; project administration, C.S.V.-S., M.A.R.-M., J.d.J.L.-S. and M.C.-L.; funding acquisition, J.d.J.L.-S. and M.C.-L. All authors have read and agreed to the published version of the manuscript.

**Funding:** This project received funding from Tecnologico de Monterrey, via the Challenge-Based Research Funding Program 2022, Project ID: E061-EHE-GI02-D-T3-E.

**Institutional Review Board Statement:** The study was conducted in accordance with the Declaration of Helsinki and approved (via exemption) by the Comité de Ética en Investigación del Instituto Tecnológico y de Estudios Superiores de Monterrey (Research Ethics Committee) (protocol code: EHE-2023-03, August 2023) for studies involving humans.

**Informed Consent Statement:** Informed consent was obtained from all subjects involved in the study.

**Data Availability Statement:** We have provided the dataset generated from the experimentation for download on the Figshare database: https://doi.org/10.6084/m9.figshare.24777084 (accessed on 28 January 2024).

**Acknowledgments:** The authors would like to acknowledge the support of Miguel Blanco Ríos and Milton Osiel Candela Leal, as well as the International IUCRC BRAIN Affiliate Site at Tecnologico de Monterrey.

**Conflicts of Interest:** The authors declare no conflicts of interest.

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
