# Peer review of "EEG and Physiological Signals Dataset from Participants during Traditional and Partially Immersive Learning Experiences in Humanities"

_data, 2024_

Round 1

Reviewer 1 Report

Comments and Suggestions for Authors

The paper is impressive and the authors' ambition in generating a dataset from an immersive learning experience in Humanities is commendable. However, I would recommend a few necessary clarifications. 

When setting up an experimental scenario, it is crucial to provide a clear justification for any actions taken or decisions made. There is no clear reason for choosing scenarios 1 to 4 in your situation. This justification could take the form of a citation or addressing practical issues that arose in generating the Dataset.

In scenario 2, on line 324, there is a "work bank" that has not been described or defined. It resides in the core of the experiment, potentially influencing the flow of the study. Also, in line 325, there is the phrase "word bank". What is their differences? Given their crucial role in the experiment, it is essential to clearly define and provide justification for their inclusion.

Comments on the Quality of English Language

Minor English editing is required

Author Response

Dear Reviewer, 

All authors thank for the time invested in this revision, and we appreciate the positive feedback you provided, as well as the requests for improving our manuscript. In the attached PDF you can find a detailed explanation on the modifications added to our paper in this new version, related to each of your comments. 

Sincerely

The authors

Reviewer 2 Report

Comments and Suggestions for Authors

In the paper entitled "EEG and physiological signals dataset from participants during traditional and partially immersive learning experiences in Humanities", the authors describe a data set that was created in the context of immersive learning experiences. The authors want to make immersive VR technologies more researchable in the field of humanities and learning. In the experiment, 24 participants, 12 in the control group and 12 in the immersive learning group were measured using questionnaires and biosensory measurements (EEG, skin conductance, etc.). The data set and the experiment are described in great detail.

I think the article is overall well crafted, but the following aspects should be addressed:
- Discussion of limitations of the dataset
- There should be another spelling correction, Manikin is often described as Manakin or page 14, instead of groups' -> grups'
- Are the Python scripts also published?
- The data set should be better described in terms of quantity in a paragraph or table

Comments on the Quality of English Language

- There should be another spelling correction, Manikin is often described as Manakin or page 14, instead of groups' -> grups'

Author Response

(The authors gave the same response as above.)

Round 2

Reviewer 2 Report

Comments and Suggestions for Authors

my concerns have been well addressed